# The Role of Erythrocyte Membrane Protein Band 4.1-like 3 in Idiopathic Pulmonary Fibrosis

**DOI:** 10.3390/ijms241210182

**Published:** 2023-06-15

**Authors:** Min Kyung Kim, Jong-Uk Lee, Sun Ju Lee, Hun Soo Chang, Jong-Sook Park, Choon-Sik Park

**Affiliations:** 1Department of Interdisciplinary, Program in Biomedical Science Major, Graduate School, Soonchunhyang University, Asan 31538, Republic of Korea; 2Department of Microbiology and BK21 Four Project, College of Medicine, Soonchunhyang University, Cheonan 31538, Republic of Korea; 3Division of Allergy and Respiratory Medicine, Department of Internal Medicine, Soonchunhyang University Bucheon Hospital, Bucheon 14584, Republic of Korea

**Keywords:** EPB41L3, lung, fibrosis, epithelium, mesenchyme, fibroblast, myofibroblast, transition

## Abstract

Novel genetic and epigenetic factors involved in the development and prognosis of idiopathic pulmonary fibrosis (IPF) have been identified. We previously observed that erythrocyte membrane protein band 4.1-like 3 (*EPB41L3*) increased in the lung fibroblasts of IPF patients. Thus, we investigated the role of *EPB41L3* in IPF by comparing the EPB41L3 mRNA and protein expression of lung fibroblast between patients with IPF and controls. We also investigated the regulation of epithelial–mesenchymal transition (EMT) in an epithelial cell line (A549) and fibroblast-to-myofibroblast transition (FMT) in a fibroblast cell line (MRC5) by overexpressing and silencing *EPB41L3*. EPB41L3 mRNA and protein levels, as measured using RT-PCR, real-time PCR, and Western blot, were significantly higher in fibroblasts derived from 14 IPF patients than in those from 10 controls. The mRNA and protein expression of EPB41L3 was upregulated during transforming growth factor-β-induced EMT and FMT. Overexpression of EPB41L3 in A549 cells using lenti-*EPB41L3* transfection suppressed the mRNA and protein expression of *N-cadherin* and *COL1A1.* Treatment with EPB41L3 siRNA upregulated the mRNA and protein expression of *N-cadherin*. Overexpression of *EPB41L3* in MRC5 cells using lenti-*EPB41L3* transfection suppressed the mRNA and protein expression of *fibronectin* and *α-SMA.* Finally, treatment with *EPB41L3* siRNA upregulated the mRNA and protein expression of *FN1, COL1A1,* and *VIM.* In conclusion, these data strongly support an inhibitory effect of *EPB41L3* on the process of fibrosis and suggest the therapeutic potential of EPB41L3 as an anti-fibrotic mediator.

## 1. Introduction

Idiopathic pulmonary fibrosis (IPF) is a chronic, progressive form of interstitial lung disease of unknown etiology characterized by progressive fibrosis and worsening lung function [1,2,3]. Although the disease course varies, IPF is usually progressive [4,5,6]. Several lines of evidence have suggested that genetic and epigenetic mechanisms play roles in the development and prognosis of IPF [7,8]. Global gene expression studies have identified several novel genes in lung tissues from IPF patients [9,10,11,12]. We also revealed that 178 of 15,020 genes are differentially expressed in lung fibroblasts from IPF patients compared with those from controls [13]. Among them, the mRNA level of erythrocyte membrane protein band 4.1-like 3 (*EPB41L3*) was 14-fold higher in IPF patients than controls.

*EPB41L3* (also called *DAL-1/4.1B*) is on 18p11.31 and is believed to enable cytoskeletal protein–membrane anchor activity, cytoskeletal rearrangements, intracellular transport, and signal transduction [14]. *EPB41L3* inhibits the progression and development of several cancers, including lung adenocarcinoma, meningioma, breast cancer, ovarian cancer, and prostate cancer [14]. Epithelial–mesenchymal transition (EMT) and fibroblast-to-myofibroblast transition (FMT) are involved in the development and progression of cancer [15]. Multiple signaling pathways play roles in EMT and FMT [14]. The extracellular matrix (ECM) is degraded during the late stages of EMT through the increased expression of proteases, such as matrix metalloproteinases (MMPs), which play a crucial role in cancer metastasis and angiogenesis [16].

Several studies have revealed the involvement of *EPB41L3* in EMT and FMT. Transforming growth factor *(TGF)*-β increased the mRNA and protein expression of EPB41L3 by the A549 cell line, and deleting *EPB41L3* attenuated the TGF-β-induced EMT in a lung cancer cell line [17]. In addition, EPB41L3 suppressed tumor cell invasion and inhibited MMP2 and MMP9 expression in an esophageal squamous cancer cell line [18]. Most studies have focused on cancer. However, few have been conducted on pulmonary fibrosis. *TGF-β1* expression is particularly high in fibrotic areas of the lungs of patients with IPF, and activated *TGF-β1* drives ECM components, such as *COL1A1 (collagen 1), FN1 (fibronectin)*, and *ACTA2* (*α-SMA*), to generate a profibrotic environment in the injured area [19,20,21,22,23]. Therefore, regulating *TGF-β1* activity with *EPB41L3* is a potential therapeutic strategy for fibrosis in patients with IPF. In the present study, the role of *EPB41L3* was investigated by comparing mRNA and protein expression between lung fibroblasts derived from patients with IPF and those of controls and by regulating the EMT of an epithelial cell line (A549) and FMT of a fibroblast cell line (MRC5) by overexpressing and silencing *EPB41L3*.

## 2. Results

### 2.1. Comparison of mRNA and Protein Expression of EPB41L3 by Fibroblasts in IPF and Control Groups

To validate the enhancement of *EPB41L3* mRNA expression by fibroblasts in our previous transcriptome chip study [13], *EPB41L3* mRNA and protein were measured using cultured primary fibroblasts obtained from the lung tissues of IPF patients (n = 14) and those of controls (n = 10). Reverse transcription polymerase chain reaction (RT-PCR) revealed 168 base-pair-sized *EPB41L3* bands in IPF and control fibroblasts (Figure 1A). Densitometry revealed that mRNA *EPB41L3* expression normalized to that of *β-actin* was 2-fold higher in IPF than control fibroblasts (0.83 (0.67–0.89) vs. 0.46 (0.37–0.52), *p* < 0.001; (Figure 1B)). Real-time PCR also revealed 8-fold higher mRNA levels of *EPB41L3* normalized to those of *β-actin* in IPF than control fibroblasts (10.59 (8.79–23.34) vs. 1.29 (0.61–1.86), *p* < 0.001; (Figure 1C)). Western blotting revealed that the levels of EPB41L3 normalized to those of β-actin were 9-fold higher in IPF than control fibroblasts (0.93 (0.75–1.15) vs. 0 (0–0.09), *p* < 0.001; Figure 1D,E)). Additionally, a positive correlation was detected between mRNA and protein EPB41L3 levels in 24 fibroblasts (r = 0.504, *p* = 0.012, Figure 1F).

### 2.2. Changes in mRNA and Protein Expression of EPB41L3 and EMT-Related Genes by A549 Cells after Stimulation with TGF-β1

After A549 cells were cultured in tissue culture medium (TCM) containing 5 ng/mL of TGF-β1 for 24, 48, and 72 h, *EPB41L3*, *N-cadherin*, *E-cadherin*, *COL1A1*, and *β-actin* were quantified using real-time PCR. A 2-fold increase in the *EPB41L3*-to-*β-actin* mRNA ratio was observed after stimulation with TGF-β1 in a time-dependent manner. Moreover, the *COL1A1*- and *N-cadherin*-to-β-actin mRNA expression ratios were 10- and 3-fold higher, respectively. In contrast, the *E-cadherin*-to-β-actin mRNA expression ratio decreased significantly after stimulation with TGF-β1 in a time-dependent manner (Figure 2A). Western blot demonstrated similar changes in protein expression (Figure 2B). The EPB41L3-, N-cadherin-, and collagen-1-to-β-actin protein expression ratios increased significantly after stimulation with TGF-β1 in a time-dependent manner, while the E-cadherin-to-β-actin ratio decreased significantly (Figure 2C).

### 2.3. Effect of Overexpressing EPB41L3 on Epithelial–Mesenchymal Transition in A549 Cells

A549 cells were transfected with lenti-*EPB41L3* or the control lentiviral vector. The mRNA levels of *EPB41L3* (normalized to *β-actin*) measured using real-time PCR increased 48 h after transfection of lenti-*EPB41L3* compared with those of the lentiviral vectors in the absence (4-fold) and presence (3-fold) of 5 ng/mL of TGF-β1 (Figure 3(A-1)). Concomitantly, 5 ng/mL of TGF-β1 increased the *N-cadherin* and *COL1A1* mRNA levels, which were significantly downregulated by transfected lenti-*EPB41L3* (Figure 3(A-2,A-4)). *E-cadherin* mRNA tended to be decreased by lenti-*EPB41L3* (Figure 3(A-3)). The protein levels of these genes changed similarly on Western blots (Figure 3B). The protein level of EPB41L3 (normalized to the β-actin level) increased 48 h after the transfection of lenti-*EPB41L3* in the absence (38-fold) and presence (26-fold) of 5 ng/mL of TGF-β1 (Figure 3(C-1)). Additionally, 5 ng/mL of TGF-β1 induced an increase in N-cadherin and collagen 1, which were completely downregulated by *EPB41L3* transfection (Figure 3(C-2,C-4)). E-cadherin was not changed by lenti-*EPB41L3* (Figure 3(C-3)). These data indicate that overexpressing EPB41L3 inhibits the EMT induced by TGF-β1.

### 2.4. EPB41L3 Gene Silencing during the EMT of A549 Cells 

Treatment with siRNA (100 nM) for 48 h decreased the mRNA expression of *EPB41L3* (normalized to the *β-actin* level) in the presence and absence of 5 ng/mL of TGF-β1 (Figure 4A). Concomitantly, EPB41L3 siRNA significantly decreased the mRNA level of *EPB41L3* to the untreated level (Figure 4A). Western blot revealed similar changes in the protein levels (Figure 4B). The protein level of EPB41L3 was decreased significantly, while the N-cadherin-to-β-actin protein expression ratio was increased significantly by *EPB41L3* siRNA in the absence or the presence of TGF-β1 (Figure 4(C-1,C-2)), while the E-cadherin-to-β-actin ratio was not significantly changed (Figure 4(C-3)).

### 2.5. Effect of EPB41L3 Overexpression and Knockdown on the FMT in MRC5 Cells

Treatment with 5 ng/mL of TGF-β1 significantly increased the mRNA expression of *FN1*, *ACTA2*, *N-cadherin*, and *COL1A1* (normalized to the *β-actin* level) of MRC-5 cells in a time-dependent manner (Figure 5(A-2–A-5)). The protein levels of these genes showed the same pattern of change (Figure 5B). The protein expression of EPB41L3, fibronectin, α-SMA, and collagen 1 increased in a time-dependent manner (Figure 5(C-1–C-4)). When *EPB41L3* was transfected into MRC5 cells with lenti-*EPB41L3*, the mRNA and protein expression of *EPB41L3* increased 48 h after transfection (Figure 6(A-1,C-1)). Concomitantly, the mRNA and protein expression of *FN1* (Figure 6(A-2,C-2)) and *ACTA2* (Figure 6(A-3,C-3)) significantly decreased after transfection with lenti-*EPB41L3* in the absence and presence of TGF-β1. Treatment with EPB41L3 siRNA (100 nM) for 48 h decreased the *EPB41L3* mRNA expression normalized to the *β-actin* level (Figure 7(A-1)). Concomitantly, the *FN1*, *COL1A1*, and *VIM* mRNA levels were significantly upregulated after treatment with *EPB41L3* siRNA (Figure 7(A-2–A-4)). Western blot demonstrated the same findings that the protein level of EPB41L3 was significantly decreased (Figure 7B,(C-1)), while the fibronectin-, collagen-1-, and vimentin-to-β-actin protein expression ratios were increased significantly by treatment with siEPB41L3 (Figure 7(C-2–C-4)). These data indicate that EPB41L3 regulates FMT.

## 3. Discussion

In the present study, the protein and mRNA expression of the *EPB41L3* gene significantly increased in lung-tissue-derived fibroblasts from IPF patients compared to those from controls. This result is in line with the transcriptomic datasets of single-cell RNA-seq demonstrating the upregulation of *EPB41L3* in the lung fibroblasts of IPF patients [24]. In A549 and MRC5, the overexpression of *EPB41L3* suppressed EMT- and FMT-related genes, while *EPB41L3* knockdown upregulated these genes. These data show for the first time that the *EPB41L3* gene may play a role in the development of pulmonary fibrosis by regulating EMT and FMT.

*EPB41L3* has been regarded as a tumor suppressor that inhibits the progression and development of several types of cancer, including lung adenocarcinoma, meningioma, breast cancer, ovarian cancer, and prostate cancer [25,26]. Loss of the 4.1B/DAL-1 protein leads to a substantial decrease in the expression of numerous EMT markers, including E-cadherin and β-catenin in lung cancer cell lines [27], which agrees with our EPB41L3 knockdown data. *EPB41L3* maintains the epithelial cell phenotype and attenuates EMT by inhibiting Snail and PI3K/Akt/Mdm2/p53 signaling [14,27,28]. In a study using 83 osteosarcoma tissues [14], the expression of Snai1/Slug/Twist1 was correlated with that of EPB41L3. Thus, *EPB41L3* may inhibit the progression of EMT by interacting with *Snail1*, *2*, and *Twist1* in patients with pulmonary fibrosis.

The mechanism underlying the increase in *EPB41L3* in IPF has not been revealed. In our previous transcriptome analysis of 12 lung fibroblast samples derived from 4 controls and 8 IPF-patients (GSE71351), strong correlations were observed between the EPB41L3 levels and those of 344 among 15,020 genes (160 positive and 184 negative correlations) (Appendix A). Among them, *ZNF678*, which is involved in transcriptional regulation, interacted with EPB41L3 in the STRING protein–protein interaction network analysis (Appendix A). Among the 99 proteins interacting with ZNF687, 33 proteins, including MYC [29], CSNK2A1 [30], CDKN2A [31], HDAC2 [32], MI1 [33], SOX2 [34], and RNF2 [35], have been revealed to be associated with fibrosis. These data indicate that EPB41L3 may be regulated through interactions with ZNF687 and interacting genes.

In addition, CpG methylation is a common mechanism of genetic downregulation. The hypermethylation of DAL-1 is strongly correlated with the loss of DAL-1 and predicts short overall survival in patients with non-small cell lung cancer (NSCLC) [36]. The hypermethylation of CpG sites in the EPB41L3 promoter region leads to loss of expression, as observed in esophageal cancer [37]. In our previous methylation study (GSE107226) using the same fibroblasts [38], 21 CpG sites were identified on the *EPB41L3* gene (5 on TSS1500, 5 on TSS200, 9 on the 5′UTR, 1 on the body, and 1 on the 3′UTR) (Appendix A). Among them, different methylation levels were detected at four CpG sites (cg15528411, cg22335490, cg27082185, and cg14075742) in the fibroblasts of IPF patients compared with the controls. Significant positive correlations were observed at one CpG site (cg27082185), and a negative correlation was observed at one CpG site (cg00027083) compared with the transcriptome levels (Appendix A and Appendix A). Although we did not conduct a functional study of these CpG sites, different CpG methylation sites may be responsible for the changes seen in the *EPB41L3* transcriptome levels.

Our study has several limitations. First, control fibroblasts were obtained from normal portions of the resected cancer specimens. The gene expression profile of fibroblasts derived from lungs in which cancer developed may be different from that of fibroblasts derived from truly normal lungs. Secondly, we used the A549 and MRC-5 cell lines in the EMT and FMT studies instead of primary lung epithelial cells or fibroblasts. Thirdly, the *EPB41L3* gene and protein levels were not measured using the samples, such as bronchoalveolar lavage or lung tissues. To reveal the exact role of EPB41L3 in IPF, the protein and gene levels of EPB41L3 should be evaluated in the lungs of the patients with IPF in terms of clinical manifestations by assessing the correlations of their levels with prognostic parameters, such as the long-term survival rate, in large number of patients. Additionally, one important limitation of our study is the lack of the functional study of EPB41L3 in IPF-related cellular processes, such as morphological changes, cell proliferation, migration, and invasion. Therefore, further functional studies are needed using EPB41L3 overexpressed and knocked cell lines.

## 4. Materials and Methods

### 4.1. Study Subjects

Lung fibroblasts and lung tissues of patients with IPF were obtained from the Biobank of Soonchunhyang University Hospital (Bucheon, South Korea) after approval of the study protocol by the Institutional Review Board of Soonchunhyang University (201910-BR-058). The lung fibroblasts were cultured from the surgical specimens of 14 patients with IPF and the normal lungs of 10 subjects, who underwent surgery to remove stage I or II lung cancer; their clinical characteristics were described previously [13]. The diagnostic criteria for IPF were based on 2011 and 2018 international consensus statements [39,40].

### 4.2. Cell Culture

A549 (human epithelial cells, ATCC-CCL-185; ATCC, Manassas, VA, USA) and MRC-5 (human fetal lung fibroblast cells; ATCC-CCL-171) were cultured in TCM consisting of RPMI-1640 culture medium (GenDEPOT, Katy, TX, USA) or MEM culture medium with 100 U/mL penicillin (GenDEPOT) and 100 μg/mL streptomycin (Gibco, Carlsbad, CA, USA). The cells were maintained in a humidified incubator with 5% CO_2_ at 37 °C. The A549 and MRC-5 cells were stimulated in TCM with 0.1% fetal bovine serum (FBS; Thermo Fisher Scientific, Rockford, IL, USA) and 5 ng/mL TGF-β1 (R&D Systems, Minneapolis, MN, USA) to induce EMT and FMT.

### 4.3. RT-PCR and Real-Time PCR Analysis of EPB41L3 mRNA Expression

Total RNA was extracted using QIAzol reagent (Qiagen, Venlo, Netherlands). Total RNA (3 μg) suspended in diethylpyrocarbonate-treated water was heated at 65 °C for 5 min with 0.5 μg of 10 mM dNTPs, which is a random hexamer (Invitrogen, Carlsbad, CA, USA), and then cooled on ice. Amplification was performed for 35 cycles (30 s at 94 °C, 30 s at 61 °C, and 30 s at 72 °C) with extension at 72 °C for 7 min. The primer sequences used were as follows: *EPB41L3*—sense 5′-TCG GAG ACT ATG ACC CAG ATG A-3′ and antisense 5′-GGT GCA AAG CGG AAC TCA CT-3′; *β-ACTIN*—sense 5′-GGA CTT CGA GCA AGA GAT GG-3′ and antisense 5′-AGC ACT GTG TTG GCG TAC AG-3′; *N-cadherin*—sense 5′-AGG GAT CAA AGC CTG GAA CA-3′ and antisense 5′-TTG GAG CCT GAG ACA CGA TT-3′; *E-cadherin*—sense 5′-CCA CCA AAG TCA CGC TGA AT-3′ and antisense 5′-GGA GTT GGG AAA TGT GAG C-3′; *COL1A1*—sense 5′-ACG TCC TGG TGA AGT TGG TC-3′ and antisense 5′-ACC AGG GAA GCC TCT CTC TC-3; *FN1*—sense 5′-ACA ACA CCG AGG TGA CTG AGA C and antisense 5′-GGA CAC AAC GAT GCT TCC TGA G; *ACTA2*—sense 5′-CTA TGC CTC TGG ACG CAC AAC T and antisense 5′-CAG ATC CAG ACG CAT GAT GGC A; *VIM*—sense 5′-AGG CAA AGC AGG AGT CCA CTG A and antisense 5′-ATC TGG CGT TCC AGG GAC TCA-3. The PCR products were separated using electrophoresis on a 1% agarose gel containing ethidium bromide in Tris-borate EDTA buffer at 100 V for 35 min and visualized under ultraviolet light. The intensities of the *EPB41L3*, *E-cadherin*, *N-cadherin*, *COL1A1*, *FN1*, *ACTA2*, and *VIM* bands were normalized to those of *β-actin*. Real-time PCR was conducted using the StepOne^TM^ Real-Time PCR System (Applied Biosystems, Foster City, CA, USA). The PCR mixture (20 μL) contained 1 μg cDNA, 10 μL of 2× Power SYBR Green PCR Master Mix (Applied Biosystems), and 1 μL of the forward and reverse primers (10 pmol each). The reaction was executed in a two-step procedure: denaturation at 95 °C for 15 s and 60 °C for 1 min, with melting at 95 °C for 15 s, 60 °C for 1 min, and 95 °C for 15 s. The results were determined using the 2^−ΔΔCT^ method (26) and are presented as fold-change normalized to *β-actin*.

### 4.4. Determination of Protein Levels Using Western Blot

Proteins were extracted from cells in lysis buffer (Thermo Fisher Scientific, Waltham, MA, USA, #89901) containing proteinase and phosphatase inhibitor cocktails (Roche Diagnostics, Basel, Switzerland, #P0044-5ML). Equal amounts of protein (40 μg) were resolved using 10% sodium dodecyl sulfate–polyacrylamide gel electrophoresis and transferred to a polyvinylidene difluoride membrane (Millipore, Billerica, MA, USA). The membranes were blocked in 5% skimmed milk and incubated for 24 h at 4 °C with the following primary antibodies: rabbit polyclonal anti-human EPB41L3 (1:2000; ProteinTech, Rosemont, IL, USA, #10719-I-AP), mouse monoclonal anti-human E-cadherin (1:1000; Invitrogen, #33-4000), mouse monoclonal anti-human N-cadherin (1:1000, Invitrogen, #33-3300), rabbit polyclonal anti-human collagen I (1:1000; Abcam, Cambridge, MA, USA, #MA1-26771), mouse monoclonal anti-human-α-SMA (1:500; Abcam, #ab7817), mouse monoclonal anti-human fibronectin (1:1000; Abcam, #ab6328), and mouse monoclonal anti-human β-actin (1:50,000; Sigma-Aldrich, St. Louis, MO, USA, #A1978). After washing several times with Tris-buffered saline containing Tween 20, the membranes were incubated with goat anti-rabbit (1:5000; GenDEPOT, #SA007-500) or goat anti-mouse immunoglobulin G (IgG) horseradish peroxidase-conjugated secondary antibody (1:5000, #SA001-500, and 1:100,000 for β-actin; GenDEPOT, #A0042-001). The membranes were analyzed using chemiluminescence (Thermo Fisher Scientific and Bio-Rad (Hercules, CA, USA)) with the ChemiDoc™ Touch Imaging System (Bio-Rad). Protein expression was normalized using β-actin as a loading control.

### 4.5. Lentiviral Transduction to Overexpress EPB41L3 in A549 and MRC5 Cells

Stable transfection was performed using the Open Reading Frame (ORF) Lentiviral Transduction Particles kit (ORIGENE, Rockville, MD, USA) according to the manufacturer’s instructions. The *EPB41L3* Human Tagged ORF (Cat: RC209736L3V; ORIGENE) and corresponding control lentiviral particles (Cat: PS100092V; ORIGENE) were used for stable transfection. Lentiviral transduction was performed when the cells reached 70% confluence. *EPB41L3* Human Tagged ORF clone-expressing lentivirus or lenti-ORF control virus particles were added to the cells at a multiplicity of infection of 5 for A549 cells and 1 for MRC5 cells, along with 8 µg/ml polybrene (Sigma-Aldrich). The cells were maintained in 5 μg/mL puromycin (Sigma-Aldrich) for approximately 10 days to establish stable cell lines. Then, mRNA and protein overexpression of EPB41L3 was determined using real-time PCR and Western blot.

### 4.6. EPB41L3 Gene Silencing Using siRNA

siRNAs against *EPB41L3* and scrambled siRNA (Bioneer, Daejeon, South Korea) were transfected into A549 and MRC5 cells using Lipofectamine 2000 (Invitrogen), according to the manufacturer’s protocol, when the cells were 70% confluent. The siRNA sequences used were as follows: si*EPB41L3*—sense 5′-GUG AAG ACG GAA ACC AUC A-3′ and antisense 5′-UGA UGG UUU CCG UCU UCA C-3′. The siRNAs (100 nM) were incubated in Opti-MEM (Gibco) solution for 5 min; 4 µL of Lipofectamine 2000 was mixed into the solution, which was then incubated for 20 min. The mixtures were added to the cells and incubated for 6 h at 37 °C. The cells were cultured in 1% FBS-TCM with or without TGF-β1 (5 ng/mL) for 48 h. Knockdown efficiencies were quantified using real-time PCR and Western blot.

### 4.7. Statistical Analysis

Data were analyzed using SPSS (ver. 20.0; SPSS, Inc., Chicago, IL, USA). The distribution of data was assessed using the Shapiro–Wilk test, and Mann–Whitney *U* test was used for the comparison of non-parametric data between two groups. Two-way ANOVA with Bonferroni’s post hoc test for multiple comparisons were used for comparison of parametric samples. Skewed data were expressed as medians with 25% and 75% quartiles, and normally distributed data were expressed as the means ± standard error of the mean. *p*-values < 0.05 were considered significant.

## 5. Conclusions

Fibroblasts from the lungs of IPF patients expressed high mRNA and protein levels of the *EPB41L3* gene compared with controls. *EPB41L3* gene and protein expression was induced during EMT and FMT. Overexpressing *EPB41L3* suppressed EMT- and FMT-related genes and proteins, while silencing *EPB41L3* upregulated them. These data demonstrate the inhibitory role of *EPB41L3* in the fibrosing process. As TGF-β levels increase in the lungs of IPF patients [41], the phenotypes of the fibroblasts may be changed by long-term stimulation with an excessive amount of TGF-β in the lungs of IPF patients. Thus, a balance between the amount of TGF-β (fibrotic factor) and EPB41L3 (anti-fibrotic factor) may determine the progress of fibrosis in patients with IPF.

## Figures and Tables

**Figure 1 ijms-24-10182-f001:**
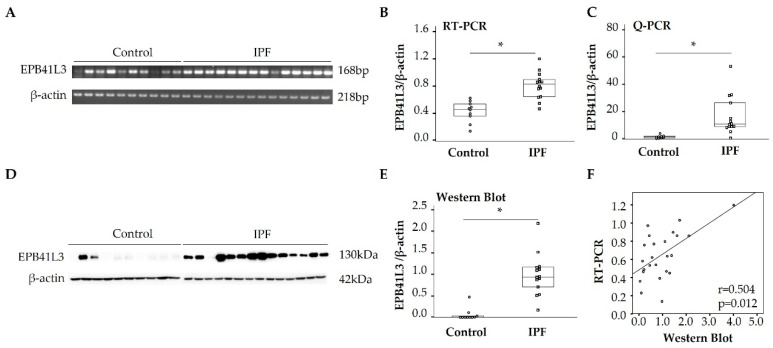
Comparison of mRNA and protein EPB41L3 levels in lung-tissue-derived fibroblasts between 14 IPF patients and 10 controls. Expression of *EPB41L3* mRNA (normalized to that of β-actin) was measured using (**A**) RT-PCR, (**B**) densitometry, and (**C**) real-time PCR (log (2^−∆∆Ct^). Protein expression of EPB41L3 (normalized to that of β-actin) was measured using (**D**) Western blot and (**E**) densitometry. The Mann–Whitney U test was performed to identify the statistical significance between IPF and control groups. Data are medians and quantiles, and * *p* < 0.01. (**F**) Correlations between the RT-PCR and Western blot band intensities was analyzed using Spearman’s correlation coefficient. (See Appendix A for RT-PCR and original Western blot images.)

**Figure 2 ijms-24-10182-f002:**
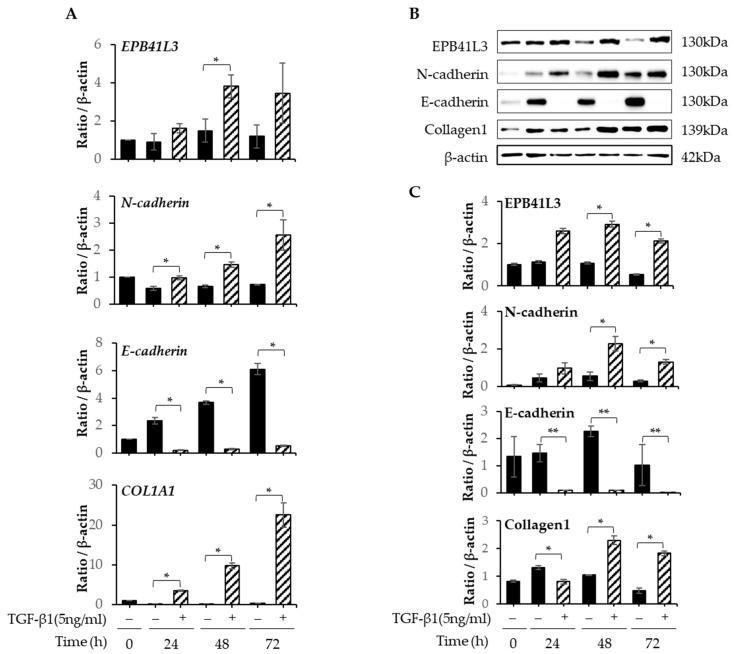
Changes in mRNA and protein of *EPB41L3* and EMT-related genes expressed by A549 cells after stimulation with TGF-β. A549 cells were cultured with or without 5 ng/mL of TGF-β1 for 24, 48, and 72 h. The *EPB41L3*-, *N-cadherin*-, *E-cadherin*-, and *COL1A1*-to-β-actin ratios were measured using (**A**) real-time PCR, (**B**) Western blot, and (**C**) densitometry of protein bands. They were normalized to β-actin and expressed as ratios. All experiments were performed separately six times, and the representative images are shown. Data are mean ± SE of the six independent experiments. Basal levels (0 h) were excluded from statistical analysis. Significant differences between groups were evaluated using two-way ANOVA with Bonferroni multiple comparisons test. * *p* < 0.05 and ** *p* < 0.01 compared to TGF-β1-stimulated cells. Black bars: untreated control group, bars with diagonals: TGF-β1-treated group. (See Appendix A for original Western blot images).

**Figure 3 ijms-24-10182-f003:**
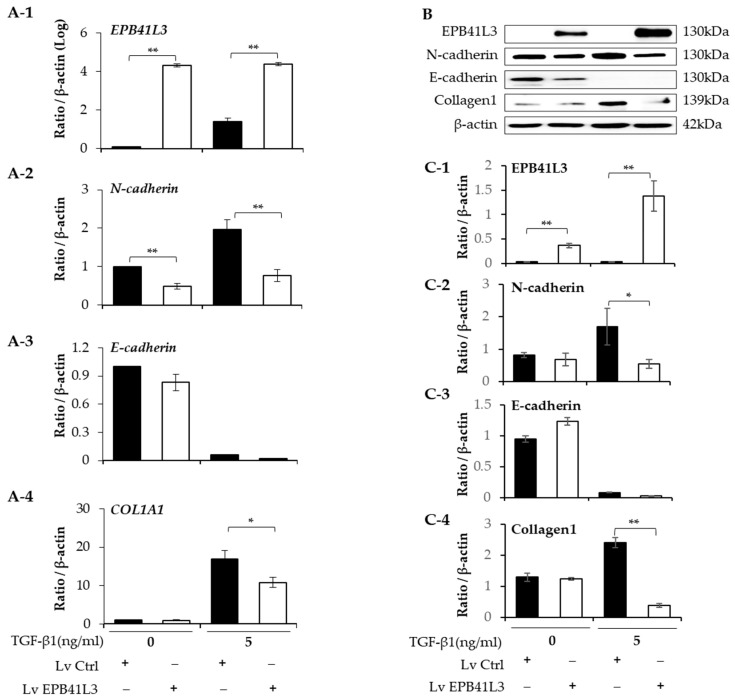
Effect of EPB41L3 transfection on EMT-related genes in the A549 cell line. Changes in EPB41L3 and EMT-related gene expression after EPB41L3 transfection in the presence and absence of 5 ng/mL of TGF-β1 for 48 h. Expression of mRNA and protein normalized to β-actin levels were measured using (**A-1**–**A-4**) real-time PCR, (**B**) Western blot, and (**C-1**–**C-4**) densitometry of protein bands. They were normalized to β-actin and expressed as ratios. Independent experiments were analyzed using densitometry. All experiments were performed separately six times, and the representative images are shown. Data are mean ± SE of 6 independent experiments. Significant differences between groups were evaluated using two-way ANOVA with Bonferroni multiple comparisons test. * *p* < 0.05 and ** *p* < 0.01. EMT: epithelial–mesenchymal transition, Lv: Lentivirus, black bar: control group, open bar: EPB41L3-overexpressing group. (See Appendix A for original Western blot images).

**Figure 4 ijms-24-10182-f004:**
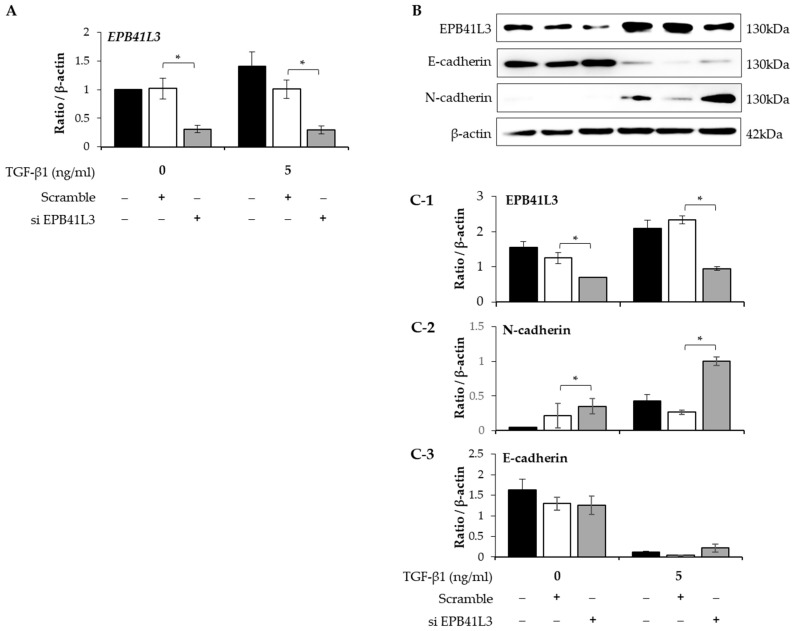
Effect of silencing EPB41L3 on EMT-related genes in A549 cells. Changes of *EPB41L3* expression in response to EPB41L3 siRNA in the presence and absence of 5 ng/mL of TGF-β1 for 48 h. Expression of mRNA and protein normalized to the β-actin level was measured using (**A**) real-time PCR, (**B**) Western blot, and (**C-1**–**C-3**) densitometry of protein bands. Independent experiments were analyzed using densitometry. They were normalized to β-actin and expressed as ratios. All experiments were performed separately six times, and the representative images are shown. Data are mean ± SE of 6 independent experiments. Significant differences between groups were evaluated using two-way ANOVA with Bonferroni multiple comparisons test. * *p* < 0.05. Black bar: control group, open bar: scramble group, Gray bar: EPB41L3-knockdown group. (See Appendix A for original Western blot images).

**Figure 5 ijms-24-10182-f005:**
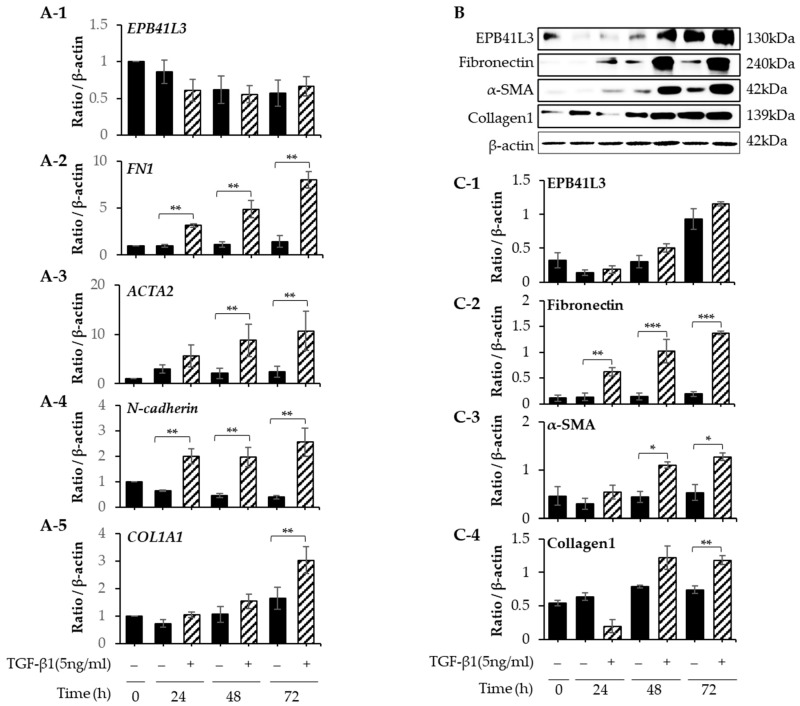
Changes in mRNA and protein levels of *EPB41L3* and FMT-related genes expressed by MRC5 cells after stimulation with TGF-β. MRC5 cells were cultured with or without 5 ng/mL of TGF-β1 for 24, 48, and 72 h. The *EPB41L3*-, *FN1*-, *ACTA2*-, *N-cadherin*-, and *COL1A1*-to-β-actin ratios were measured using (**A-1**–**A-5**) real-time PCR, (**B**) Western blot and (**C-1**–**C-4**) densitometry. Independent experiments were analyzed using densitometry. They were normalized to β-actin and expressed as ratios. All experiments were performed separately six times, and the representative images are shown. Data are mean ± SE of 6 independent experiments. Basal levels (0 h) were excluded from statistical analysis. Significant differences between groups were evaluated using two-way ANOVA with Bonferroni multiple comparisons test. * *p* < 0.05, ** *p* < 0.01 and *** *p* < 0.001 compared to TGF-β1-stimulated cells. Black bar: control group, open bar: EPB41L3-overexpressing group. (See Appendix A for original Western blot images).

**Figure 6 ijms-24-10182-f006:**
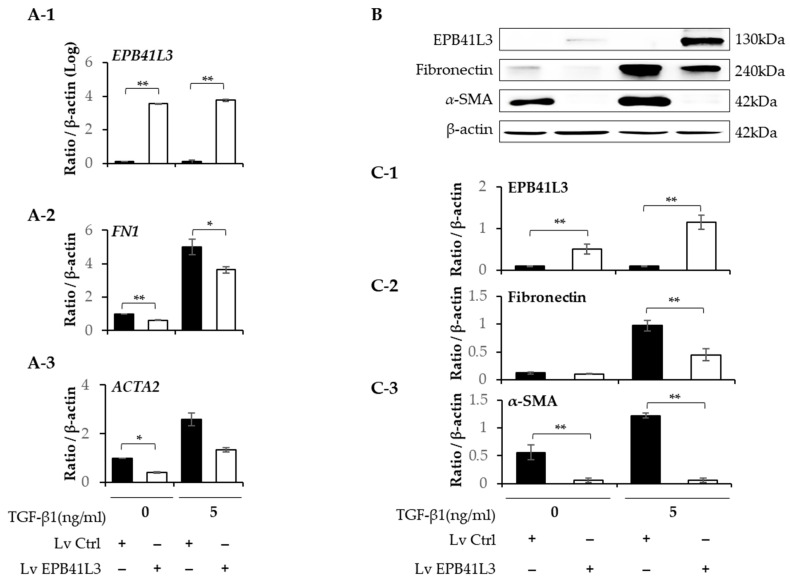
Effect of EPB41L3 transfection on FMT-related genes in the MRC5 cell line. Changes in the expression of EPB41L3 and FMT-related genes in response to EPB41L3 transfection in the presence and absence of 5 ng/mL of TGF-β1 for 48 h. Expression of mRNA and protein normalized to the β-actin level were measured using (**A-1**–**A-3**) real-time PCR, (**B**) Western blot, and (**C-1**–**C-3**) densitometry of protein bands. They were normalized to β-actin and expressed as ratios. Independent experiments were analyzed using densitometry. All experiments separately were performed six times, and the representative images are shown. Data are mean ± SE of 6 independent experiments. Significant differences between groups were evaluated using two-way ANOVA with Bonferroni multiple comparisons test. * *p* < 0.05 and ** *p* < 0.01. Lv: Lentivirus, black bar: control group, open bar: EPB41L3-overexpressing group. (See Appendix A for original Western blot images).

**Figure 7 ijms-24-10182-f007:**
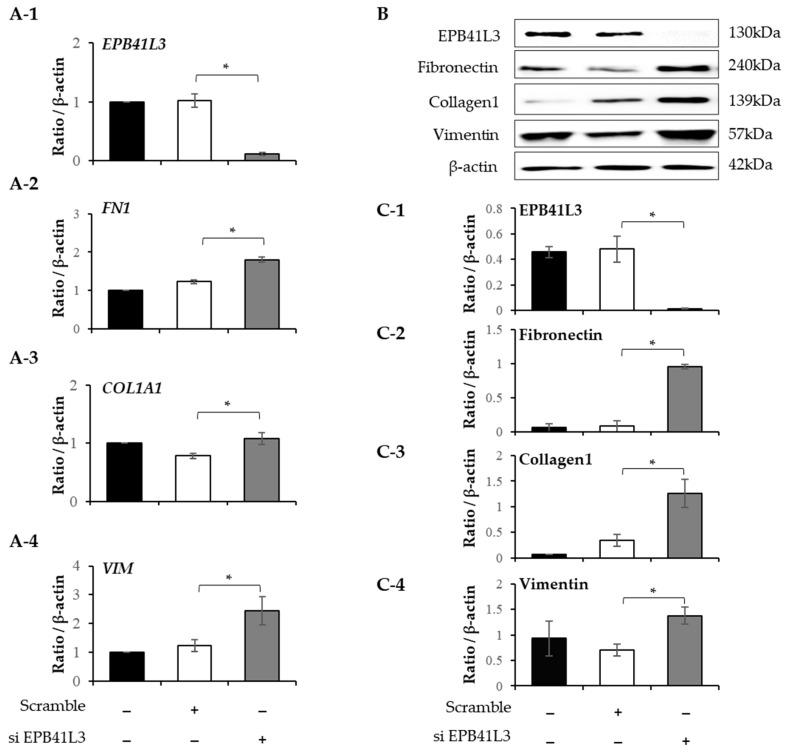
Effect of silencing EPB41L3 on FMT-related genes in MRC5 cells. Changes in the expression of EPB41L3 and FMT-related genes in response to EPB41L3 siRNA. Expression of mRNA and protein were measured using (**A-1**–**A-4**) real-time PCR, (**B**) Western blot, and (**C-1**–**C-4**) densitometry of protein bands. Independent experiments were analyzed using densitometry. They were normalized to β-actin and expressed as ratios. All experiments separately were performed six times, and the representative images are shown. Data are mean ± SE of 6 independent experiments. Significant differences between groups were evaluated using two-way ANOVA with Bonferroni multiple comparisons test. * *p* < 0.05. Black bar: control group, open bar: scramble group, gray bar: EPB41L3-knockdown group. (See Appendix A for original Western blot images).

## Data Availability

The data are freely available.

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
