# Peer review of "The Role of Erythrocyte Membrane Protein Band 4.1-like 3 in Idiopathic Pulmonary Fibrosis"

_ijms, 2023, doi:10.3390/ijms241210182_

Round 1

Reviewer 1 Report

Although not very original, this manuscript brings new data on the mechanism of myofibroblast differentiation during IPF. Result description is convincing and discussion is clear.  With several improvements to english language it may be accepted as it stands.

Several points should be improved, e.g.Inroduction lines 3-4: "Although the disease course varies. IPF is usually progressive." ( no need to add "in cases of irreversible pulmonary fibrosis."

Author Response

Title: The Role of Erythrocyte Membrane Protein Band 4.1-Like 3 in Idiopathic Pulmonary Fibrosis

Manuscript ID: ijms-2413057

Editorial Department,  “International Journal of Molecular Sciences”

Dear Editors,

I wish to express my sincere appreciation to the reviewers for their constructive comments on the manuscript. As recommended, we have attempted to address the comments of reviewers in a point by point response.

We believe that the revision has increased the overall quality of the manuscript and hope that our results fulfill the high standards of International Journal of Molecular Sciences

With my best regards,

Choon-Sik Park, M.D., Ph.D.,

Division of Allergy and Respiratory Medicine,

Department of Internal Medicine,

Soonchunhyang University Bucheon Hospital,

1174, Jung-Dong, Wonmi-Ku, Bucheon, Kyeonggi-Do, 420-020, Korea

Tel: 82-32-621-5105, Fax: 82-32-621-5023,

Email: Choon-Sik Park; [email protected]

Response to Reviewer 1 Comments

Point1: Several points should be improved, e.g. Introduction lines 3-4: "Although the disease course varies. IPF is usually progressive." (no need to add "in cases of irreversible pulmonary fibrosis."

Response1: Thanks for your comment. According to reviewer comment, we deleted the sentence "in cases of irreversible pulmonary fibrosis." as below:

Previously sentence:

Idiopathic pulmonary fibrosis (IPF) is a chronic, progressive form of interstitial lung disease of unknown etiology characterized by progressive fibrosis and worsening lung function [1, 2]. Although the disease course varies, IPF is usually progressive in cases of irreversible pulmonary fibrosis [3, 4].

Revised sentence:

Idiopathic pulmonary fibrosis (IPF) is a chronic, progressive form of interstitial lung disease of unknown etiology characterized by progressive fibrosis and worsening lung function [1, 2]. Although the disease course varies, IPF is usually progressive [3, 4].

Reviewer 2 Report

The manuscript entitled " The Role of Erythrocyte Membrane Protein Band 4.1-Like 3 in Idiopathic Pulmonary Fibrosis " in which the authors examined the role of EPB41L3 in IPF by comparing EPB41L3 mRNA and protein expression of lung fibroblast between patients with IPF and controls. Also, they investigated the regulation of the epithelial-mesenchymal transition (EMT) in an epithelial cell line (A549), and the fibroblast-to-myofibroblast transition (FMT) in a fibroblast cell line (MRC5). They found an inhibitory effect of EPB41L3 on the process of fibrosis and suggest therapeutic potential of EPB41L3 as an anti-fibrotic mediator

The work is understandable and the topic is important. The scientific narrative is well structured and flows naturally from one idea to the next. The results are interesting although the study had some limitations.

However, this paper suffers from some shortcomings that if modified would make the manuscript very suitable for publication in International Journal of Molecular Sciences.

Shortcomings:

1-      Please add the amount of used proteins in western blot part in methods section.

2-      If the target protein levels in western blot were normalized to β-actin as a housekeeping protein, please write that in figure legend of western blot data.

3-      In figures of western blot data, did the authors use the same membrane to check the different proteins in each experiment? please clarify?

4-      I recommend to collect the western blot figures in whole membranes of one experiment in a single figure and the same for other experiments to have one supplementary PDF file for original western blot data.

Author Response

Title: The Role of Erythrocyte Membrane Protein Band 4.1-Like 3 in Idiopathic Pulmonary Fibrosis

Manuscript ID: ijms-2413057

Editorial Department,  “International Journal of Molecular Sciences”

Dear Editors,

I wish to express my sincere appreciation to the reviewers for their constructive comments on the manuscript. As recommended, we have attempted to address the comments of reviewers in a point by point response.

We believe that the revision has increased the overall quality of the manuscript and hope that our results fulfill the high standards of International Journal of Molecular Sciences

With my best regards,

Choon-Sik Park, M.D., Ph.D.,

Division of Allergy and Respiratory Medicine,

Department of Internal Medicine,

Soonchunhyang University Bucheon Hospital,

1174, Jung-Dong, Wonmi-Ku, Bucheon, Kyeonggi-Do, 420-020, Korea

Tel: 82-32-621-5105, Fax: 82-32-621-5023,

Email: Choon-Sik Park; [email protected]

Response to Reviewer 2 Comments

Point 1: Please add the amount of used proteins in western blot part in methods section.

Response 1: Thanks for your comment. We added the amount of proteins used in western blot as below:

Proteins were extracted from cells in lysis buffer (Thermo Fisher Scientific, Waltham, MA, USA) containing proteinase and phosphatase inhibitor cocktails (Roche Diagnostics, Basel, Switzerland). Equal amounts of protein (40 μg) were resolved by 10% sodium dodecyl sulfate-polyacrylamide gel electrophoresis and transferred to a polyvinylidene difluoride membrane (Millipore, Billerica, MA, USA).

Point 2: If the target protein levels in western blot were normalized to β-actin as a housekeeping protein, please write that in figure legend of western blot data.

Response 2: Thanks for your comment. We changed the sentence in figure legend of Figure 7 as follows:

Figure 7. Effect of silencing EPB41L3 on FMT-related genes in MRC5 cells

Changes in the expression of EPB41L3 and FMT-related genes in response to EPB41L3 siRNA. Expression of mRNA and protein were measured by (A) real-time PCR, (B) western blot, and (C) densitometry of protein bands. They were normalized to β-actin and expressed as ratio. Independent experiments were analyzed using densitometry. All experiments were performed six times and the representative image was shown. Data are mean ± SE of 6 independent experiments. Significant differences between groups were evaluated by two-way ANOVA with Bonferroni’s multiple comparisons test for pairwise comparisons. * p <0.05. Black bar: control group, open bar: scramble group, gray bar: EPB41L3-knockdown group. (See Figure S7 for and RT-PCR original Western blot images).

Point 3: In figures of western blot data, did the authors use the same membrane to check the different proteins in each experiment? please clarify?

Response 3: Thanks for the pointing it out. We added the sentence to clarify the sentence in figure legend according to the reviewer comment, and original western blot images were presented the supplementary PDF file.

Point 4: I recommend to collect the western blot figures in whole membranes of one experiment in a single figure and the same for other experiments to have one supplementary PDF file for original western blot data.

Response 4: Thanks for your comment. We collected the original western blot figures and presented the supplementary PDF file.

Reviewer 3 Report

The authors present a relevant study and it will surely be of great interest to various research groups. However, after the review, I will focus on the results with these series of points:
1) Figure 1 indicates the statistical test applied for the correlation, so it would also be appropriate to indicate the Mann-Whitney U test.
2) Figure 1, do not repeat the statistical values that were located at the bottom of the figure (r = 0.504, p = 0.012) if these are already integrated in the F graph.
3) Figure 2, the comparisons of the graphs that are integrated into A and C, show 7 groups for the contrast located at 0, 24, 48, and 72 h. The authors explain that they applied t-tests, but the corresponding test is a one-way ANOVA and, correspondingly, make multiple comparisons between all the average values or, failing that, select between the pairs of interest (negative vs. positive). However, even with this context I still have more doubts, because the 4 times are also a reason to compare (0, 24, 48, and 72) and in this sense, it would be a second factor. In addition, the level corresponding to time 0 would also require its corresponding group (+). So your analysis design is a two-factor model.
4) In the same sense, it is also important to make it clear whether the records are independent (replicas) or pseudo-replicas, information that is not indicated in the Methods or in the Statistical Analysis section.
5) In Figure 2, it is not indicated what represents the bars in black and those with diagonals.
6) Figure 3, also requires the information of the test that was applied, and again, if in case they are t-tests to compare in 0 negative vs positive (black bar vs. empty bar), then the graphs of 0 should not be united those of 5. Graphically the message is that they are comparing the four groups, so the type of statistical test must be defined very well and indicate the corresponding information as well as the way in which it is graphed.
7) Figure 4, at the end of their information they delimit that there are 6 independent experiments, so what happens in the previous results (figure 1-3)? Here in this figure 4, if there is independence in the records, it would be appropriate to apply a two-way ANOVA with a factor with two levels (0.5) and another factor with dark, white, and gray bars.
8) In Figures 5, 6, and 7, the authors also have to clarify under the same argument what they explain in the previous figures.
9) Review the guidelines for references, since the titles of the articles were placed in italics, which is contrary to what the editorial standards of the journal indicate. In addition, the titles of the journals should be in italics. Also, there are more errors, since some names are abbreviated and others are not (check all references and adjust).

Kind regards,

Requires a style review.

Author Response

Title: The Role of Erythrocyte Membrane Protein Band 4.1-Like 3 in Idiopathic Pulmonary Fibrosis

Manuscript ID: ijms-2413057

Editorial Department,  “International Journal of Molecular Sciences”

Dear Editors,

I wish to express my sincere appreciation to the reviewers for their constructive comments on the manuscript. As recommended, we have attempted to address the comments of reviewers in a point by point response.

We believe that the revision has increased the overall quality of the manuscript and hope that our results fulfill the high standards of International Journal of Molecular Sciences

With my best regards,

Choon-Sik Park, M.D., Ph.D.,

Division of Allergy and Respiratory Medicine,

Department of Internal Medicine,

Soonchunhyang University Bucheon Hospital,

1174, Jung-Dong, Wonmi-Ku, Bucheon, Kyeonggi-Do, 420-020, Korea

Tel: 82-32-621-5105, Fax: 82-32-621-5023,

Email: Choon-Sik Park; [email protected]

Response to Reviewer 3 Comments

Point 1: Figure 1 indicates the statistical test applied for the correlation, so it would also be appropriate to indicate the Mann-Whitney U test.

Response 1: Thanks for your comment. We changed the figure legend to clarify statistical method as below:

Figure 1. Comparison of mRNA and protein EPB41L3 levels in lung tissue-derived fibroblasts between 14 IPF patients and 10 controls.

Expression of EPB41L3 mRNA (normalized to that of β-actin) was measured using (A) RT-PCR, (B) densitometry and (C) real-time PCR (log (2-△△Ct). Protein expression of EPB41L3 (normalized to that of β-actin) was measured using (D) western blot and (E) densitometry. The Mann–Whitney U test was performed to identify the statistical significance between IPF and control. Data are medians and quantiles, *p < 0.01. (F) Correlations between the RT-PCR and western blot band intensities was analyzed using Spearman’s correlation coefficient. (See Figure S1 for RT-PCR and original Western blot images).

Point 2: Figure 1, do not repeat the statistical values that were located at the bottom of the figure (r = 0.504, p = 0.012) if these are already integrated in the F graph.

Response 2: Thanks for your comment. We deleted the statistical values in figure legend as following the figure legend:

Previously figure legend:

Figure 1. Comparison of mRNA and protein EPB41L3 levels in lung tissue-derived fibroblasts between 14 IPF patients and 10 controls.

Expression of EPB41L3 mRNA (normalized to that of β-actin) was measured using (A) RT-PCR, (B) densitometry and (C) real-time PCR (log (2-△△Ct). Protein expression of EPB41L3 (normalized to that of β-actin) was measured using (D) western blot and (E) densitometry. The Mann–Whitney U test was performed to identify the statistical significance between IPF and control. Data are medians and quantiles, *p < 0.01. (F) Correlations between the RT-PCR and western blot band intensities was analyzed using Spearman’s correlation coefficient (r = 0.504, p = 0.012). Data are medians and quantiles, *p < 0.01. (See Figure S1 for and RT-PCR original Western blot images).

Revised figure legend:

Figure 1. Comparison of mRNA and protein EPB41L3 levels in lung tissue-derived fibroblasts between 14 IPF patients and 10 controls.

Expression of EPB41L3 mRNA (normalized to that of β-actin) was measured using (A) RT-PCR, (B) densitometry and (C) real-time PCR (log (2-△△Ct). Protein expression of EPB41L3 (normalized to that of β-actin) was measured using (D) western blot and (E) densitometry. The Mann–Whitney U test was performed to identify the statistical significance between IPF and control. Data are medians and quantiles, *p < 0.01. (F) Correlations between the RT-PCR and western blot band intensities was analyzed using Spearman’s correlation coefficient. Data are medians and quantiles, *p < 0.01. (See Figure S1 for and RT-PCR original Western blot images).

Point 3: Figure 2, the comparisons of the graphs that are integrated into A and C, show 7 groups for the contrast located at 0, 24, 48, and 72 h. The authors explain that they applied t-tests, but the corresponding test is a one-way ANOVA and, correspondingly, make multiple comparisons between all the average values or, failing that, select between the pairs of interest (negative vs. positive). However, even with this context I still have more doubts, because the 4 times are also a reason to compare (0, 24, 48, and 72) and in this sense, it would be a second factor. In addition, the level corresponding to time 0 would also require its corresponding group (+). So your analysis design is a two-factor model.

Response 3: Thanks for the pointing it out. All experiments were performed six times, and we reanalyzed data using a two-way ANOVA according to reviewer comment. Also, we revised the statistical method as below:

Previously sentence 1:

Figure 2. Changes in mRNA and protein expression of EPB41L3 and EMT-related genes expressed by A549 cells after stimulation with TGF-β.

A549 cells were cultured with or without 5 ng/mL TGF-β1 for 24, 48, and 72 h. The EPB41L3, N-cadherin, E-cadherin, and COL1A1 to β-actin ratios were measured using (A) real-time PCR, (B) western blot, and (C) densitometry of protein bands.  Data are mean ± SE of six independent experiments. EMT: epithelial-mesenchymal transition. * p < 0.05 and ** p <0.01.

Revised sentence 1:

Figure 2. Changes in mRNA and protein of EPB41L3 and EMT-related genes expressed by A549 cells after stimulation with TGF-β.

A549 cells were cultured with or without 5 ng/mL TGF-β1 for 24, 48, and 72 h. The EPB41L3, N-cadherin, E-cadherin, and COL1A1 to β-actin ratios were measured using (A) real-time PCR, (B) western blot, and (C) densitometry of protein bands. They were normalized to β-actin and expressed as ratio. All experiments were separately performed six times and the representative image was shown. Data are mean ± SE of the six independent experiments. Basal levels (0 hour) were excluded to statistical analysis. Significant differences between groups were evaluated by two-way ANOVA with Bonferroni’s for multiple comparisons test. * p < 0.05 and ** p <0.01 compared to TGF- β1 stimulated cells. Black bars: untreated control group, Bars with diagonals: TGF-β1 - treated group. (See Figure S2 for original Western blot images).

Previously sentence 2:

4.7. Statistical analysis

Data were analyzed using SPSS software (ver. 20.0; SPSS Inc., Chicago, IL, USA). The data distribution was assessed using the Shapiro-Wilk test. Student’s t-test or the Mann–Whitney U test was used to analyze continuous data. Skewed data are expressed as medians with 25% and 75% quartiles, and normally distributed data are expressed as mean ± standard error. Correlations between mRNA and protein EPB41L3 levels were analyzed using Spearman’s correlation coefficient analysis. A p value < 0.05 was considered significant.

Revised sentence 2:

4.7. Statistical analysis

Data were analyzed using SPSS (ver. 20.0; SPSS, Inc., Chicago, IL, USA). The distribution of data was assessed using the Shapiro–Wilk test. Mann–Whitney test for the comparison of non-parametric data between two groups.  Two-way ANOVA with Bonferroni’s post hoc test for multiple comparisons were used for comparison of parametric samples. Skewed data were expressed as medians with 25% and 75% quartiles and normally distributed data were expressed as the means ± standard error of the mean. P-values < 0.05 were considered significant.

Point 4: In the same sense, it is also important to make it clear whether the records are independent (replicas) or pseudo-replicas, information that is not indicated in the Methods or in the Statistical Analysis section.

Response 4: We added the sentence to clarify the sentence in figure legend according to the reviewer comment:

All experiments were separately performed six times and the representative image was shown.

Point 5: In Figure 2, it is not indicated what represents the bars in black and those with diagonals.

Response 5: Thanks for the pointing it out. We added the explaining what represents the bars in black and those with diagonals.

Figure 2. Changes in mRNA and protein of EPB41L3 and EMT-related genes expressed by A549 cells after stimulation with TGF-β.

A549 cells were cultured with or without 5 ng/mL TGF-β1 for 24, 48, and 72 h. The EPB41L3, N-cadherin, E-cadherin, and COL1A1 to β-actin ratios were measured using (A) real-time PCR, (B) western blot, and (C) densitometry of protein bands. They were normalized to β-actin and expressed as ratio. All experiments were separately performed six times and the representative image was shown. Data are mean ± SE of the six independent experiments. Basal levels (0 hour) were excluded to statistical analysis. Significant differences between groups were evaluated by two-way ANOVA with Bonferroni’s for multiple comparisons test. * p < 0.05 and ** p <0.01 compared to TGF- β1 stimulated cells. Black bars: untreated control group, Bars with diagonals: TGF-β1 - treated group. (See Figure S2 for original Western blot images).

Point 6: Figure 3, also requires the information of the test that was applied, and again, if in case they are t-tests to compare in 0 negative vs positive (black bar vs. empty bar), then the graphs of 0 should not be united those of 5. Graphically the message is that they are comparing the four groups, so the type of statistical test must be defined very well and indicate the corresponding information as well as the way in which it is graphed.

Response 6: Thanks for your comment. We added the explaining what represents the empty and black bar, and revised the statistical as below:

Revised figure legend:

Figure 3. Effect of EPB41L3 transfection on EMT-related genes in the A549 cell line.

Changes in EPB41L3 and EMT-related gene expression after EPB41L3 transfection in the presence and absence of 5 ng/mL TGF-β1 for 48 h. Expression of mRNA and protein normalized to β-actin levels were measured by (A) real-time PCR, (B) western blot, and (C) densitometry of protein bands. They were normalized to β-actin and expressed as ratio. Independent experiments were analyzed using densitometry. All experiments separately were performed six times and the representative image was shown. Data are mean ± SE of 6 independent experiments.  Significant differences between groups were evaluated by two-way ANOVA with Bonferroni’s for multiple comparisons test. * p <0.05 and ** p <0.01, EMT: epithelial-mesenchymal transition, Lv: Lentivirus, black bar: control group, open bar: EPB41L3-overexpressing group. (See Figure S3 for original Western blot images).

Revised statistical method:

4.7. Statistical analysis

Data were analyzed using SPSS (ver. 20.0; SPSS, Inc., Chicago, IL, USA). The distribution of data was assessed using the Shapiro–Wilk test. Mann–Whitney test for the comparison of non-parametric data between two groups. Two-way ANOVA with Bonferroni’s post hoc test for multiple comparisons were used for comparison of parametric samples. Skewed data were expressed as medians with 25% and 75% quartiles and normally distributed data were expressed as the means ± standard error of the mean. P-values < 0.05 were considered significant.

Point 7: Figure 4, at the end of their information they delimit that there are 6 independent experiments, so what happens in the previous results (figure 1-3)? Here in this figure 4, if there is independence in the records, it would be appropriate to apply a two-way ANOVA with a factor with two levels (0.5) and another factor with dark, white, and gray bars.

Response 7: Thanks for the pointing it out. We added a statistical method to the legend of Figure 4 as below:

Figure 4. Effect of silencing EPB41L3 on EMT-related genes in A549 cells

Changes of EPB41L3 expression in response to EPB41L3 siRNA in the presence and absence of 5 ng/mL TGF-β1 for 48 h. Expression of mRNA and protein normalized to the β-actin level was measured using (A) real-time PCR, (B) western blot, and (C) densitometry of protein bands. Independent experiments were analyzed using densitometry. They were normalized to β-actin and expressed as ratio. All experiments were separately performed six times and the representative image was shown. Data are mean ± SE of 6 independent experiments. Significant differences between groups were evaluated by two-way ANOVA with Bonferroni’s for multiple comparisons test. * p <0.05, Black bar: control group, open bar: scramble group, Gray bar: EPB41L3-Knockdown group. (See Figure S4 for original Western blot images).

Point 8: In Figures 5, 6, and 7, the authors also have to clarify under the same argument what they explain in the previous figures.

Response 8: We totally agree with the reviewer's comments. For clarify the explain in all figure, we added a brief statistical method at figure legend as below:

Figure 5. Changes in mRNA and protein levels of EPB41L3 and FMT-related genes expressed by MRC5 cells after stimulation with TGF-β

MRC5 cells were cultured with or without 5 ng/mL TGF-β1 for 24, 48, and 72 h.

The EPB41L3, FN1, ACTA2, N-cadherin, and COL1A1 to β-actin ratios were measured using (A) real-time PCR, (B) western blot and (C) densitometry. Independent experiments were analyzed using densitometry. They were normalized to β-actin and expressed as ratio. All experiments were separately performed six times and the representative image was shown. Data are mean ± SE of 6 independent experiments. Basal levels (0 hour) were excluded to statistical analysis. Significant differences between groups were evaluated by two-way ANOVA with Bonferroni’s for multiple comparisons test. * p <0.05 and ** p <0.01, Black bar: control group, open bar: EPB41L3-overexpressing group. (See Figure S5 for original Western blot images).

Figure 6. Effect of EPB41L3 transfection on FMT-related genes in the MRC5 cell line.

Changes in the expression of EPB41L3 and FMT-related genes in response to EPB41L3 transfection in the presence and absence of 5 ng/mL TGF-β1 for 48 h. Expression of mRNA and protein normalized to the β-actin level were measured using (A) real-time PCR, (B) western blot, and (C) densitometry of protein bands. They were normalized to β-actin and expressed as ratio. Independent experiments were analyzed using densitometry. All experiments separately were performed six times and the representative image was shown. Data are mean ± SE of 6 independent experiments. Significant differences between groups were evaluated by two-way ANOVA with Bonferroni’s for multiple comparisons test. * p <0.05 and ** p <0.01. Lv: Lentivirus, black bar: Control group, open bar: EPB41L3-overexpressing group. (See Figure S6 for original Western blot images).

Figure 7. Effect of silencing EPB41L3 on FMT-related genes in MRC5 cells

Changes in the expression of EPB41L3 and FMT-related genes in response to EPB41L3 siRNA. Expression of mRNA and protein were measured by (A) real-time PCR, (B) western blot, and (C) densitometry of protein bands. Independent experiments were analyzed using densitometry. They were normalized to β-actin and expressed as ratio. All experiments separately were performed six times and the representative image was shown. Data are mean ± SE of 6 independent experiments. Significant differences between groups were evaluated by two-way ANOVA with Bonferroni’s for multiple comparisons test. * p <0.05. Black bar: control group, open bar: scramble group, gray bar: EPB41L3-Knockdown group. (See Figure S7 for original Western blot images).

Point 9: Review the guidelines for references, since the titles of the articles were placed in italics, which is contrary to what the editorial standards of the journal indicate. In addition, the titles of the journals should be in italics. Also, there are more errors, since some names are abbreviated and others are not (check all references and adjust).

Response 9: Thanks for the pointing it out. we revised the manuscript including title and reference according to guidelines of IJMS.

Reviewer 4 Report

In this research, Min Kyung Kim et al. demonstrated that EPB41L3 expression levels are overexpressed in lung fibroblasts from patients with PFI compared to the respective controls. Furthermore, they showed that TGF-β stimulation of A549 and MRC5 cells promotes EPB41L3 expression. They also show that overexpression of EPB41L3 on A549 and MRC5 cells by lenti-EPB41L3 transfection suppressed mRNA and protein expression of N-cadherin, COL1A1, fibronectin and α-SMA, whereas silencing of EPB41L3 on A549 and MRC5 cells by EPB41L3 siRNA treatment increased mRNA and protein expression of N-cadherin, FN1, COL1A1 and VIM. This suggests that EPB41L3 may play an essential role in the development of IPF by participating in the epithelial-mesenchymal transition (EMT) and fibroblast-to-myofibroblast transition (FMT).

The article has merit and presents some interesting data, but I think it is hampered by some critical weaknesses that need to be addressed.

1.       The authors should make sure to provide the brand and catalogue number of the reagents used for reproducibility purposes, for example in the materials and methods section specifically in Determination of protein levels by western blot the authors mention that Proteins were extracted from cells in lysis buffer but there is no mention of the buffer used, nor the catalogue number.

2.       The article is difficult to read and a clinician may not find the clinical outcome of his research, the description of the results is often very repetitive, especially when it is mentioned that mRNA and protein normalized with β-actin, a tip that can be taken into account is that the authors should report if there was a significant increase in mRNA and protein expression in relation to any treatment with respect to the control, and in the methodology express that the values of mRNA and protein expression were normalized using β-actin as a loading control, this could improve the text presented in the results.

3.       The authors demonstrate that overexpression of EPB41L3 in A549 and MRC5 cells by lenti-EPB41L3 transfection suppresses mRNA and protein expression associated with epithelial-mesenchymal transition (EMT) and fibroblast-myofibroblast transition (FMT). Whereas EPB41L3 silencing in A549 and MRC5 cells by EPB41L3 siRNA treatment increased the expression of mRNA and proteins associated with epithelial-mesenchymal transition (EMT) and fibroblast-myofibroblast transition (FMT). However, a major limitation of the study is that the functional role of EPB41L3 in IPF-related cellular processes associated with epithelial-mesenchymal transition (EMT) and fibroblast-myofibroblast transition (FMT), such as morphological changes, cell proliferation, migration, and invasion, was not fully assessed.

Author Response

Title: The Role of Erythrocyte Membrane Protein Band 4.1-Like 3 in Idiopathic Pulmonary Fibrosis

Manuscript ID: ijms-2413057

Editorial Department,  “International Journal of Molecular Sciences”

Dear Editors,

I wish to express my sincere appreciation to the reviewers for their constructive comments on the manuscript. As recommended, we have attempted to address the comments of reviewers in a point by point response.

We believe that the revision has increased the overall quality of the manuscript and hope that our results fulfill the high standards of International Journal of Molecular Sciences

With my best regards,

Choon-Sik Park, M.D., Ph.D.,

Division of Allergy and Respiratory Medicine,

Department of Internal Medicine,

Soonchunhyang University Bucheon Hospital,

1174, Jung-Dong, Wonmi-Ku, Bucheon, Kyeonggi-Do, 420-020, Korea

Tel: 82-32-621-5105, Fax: 82-32-621-5023,

Email: Choon-Sik Park; [email protected]

Response to Reviewer 4 Comments

Point 1: The authors should make sure to provide the brand and catalogue number of the reagents used for reproducibility purposes, for example in the materials and methods section specifically in Determination of protein levels by western blot the authors mention that Proteins were extracted from cells in lysis buffer but there is no mention of the buffer used, nor the catalogue number.

Response 1: Thanks for your comment. We added the catalogue number including lysis buffer and antibodies as follows:

4.4 Determination of protein levels by western blot

Proteins were extracted from cells in lysis buffer (Thermo Fisher Scientific, Waltham, MA, USA, #89901) containing proteinase and phosphatase inhibitor cocktails (Roche Diagnostics, Basel, Switzerland, #P0044-5ML). Equal amounts of protein (40 μg) were resolved by 10% sodium dodecyl sulfate-polyacrylamide gel electrophoresis and transferred to a polyvinylidene difluoride membrane (Millipore, Billerica, MA, USA). The membranes were blocked in 5% skimmed milk and incubated for 24 h at 4°C with the following primary antibodies: rabbit polyclonal anti-human EPB41L3 (1:2,000; Proteintech, Rosemont, IL, USA, #10719-I-AP), mouse monoclonal anti-human E-cadherin (1:1,000; Invitrogen, #33-4000), mouse monoclonal anti-human N-cadherin (1:1,000, Invitrogen, #33-3300), rabbit polyclonal anti-human collagen I (1:1,000; Abcam, Cambridge, MA, USA, #MA1-26771), mouse monoclonal anti-human-α-SMA (1:500; Abcam, #ab7817), mouse monoclonal anti-human fibronectin (1:1,000; Abcam, #ab6328), and mouse monoclonal anti-human β-actin (1:50,000; Sigma-Aldrich, St. Louis, MO, USA, #A1978). After washing several times with Tris-buffered saline containing Tween 20, the membranes were incubated with goat anti-rabbit (1:5,000; GenDEPOT, #SA007-500) or goat anti-mouse immunoglobulin G (IgG) horseradish peroxidase-conjugated secondary antibody (1:5,000, #SA001-500 and 1:100,000 for β-actin; GenDEPOT, #A0042-001). The membranes were analyzed using chemiluminescence [Thermo Fisher Scientific and Bio-Rad (Hercules, CA, USA)] with the ChemiDoc™ Touch Imaging System (Bio-Rad).

Point 2: The article is difficult to read and a clinician may not find the clinical outcome of his research, the description of the results is often very repetitive, especially when it is mentioned that mRNA and protein normalized with β-actin, a tip that can be taken into account is that the authors should report if there was a significant increase in mRNA and protein expression in relation to any treatment with respect to the control, and in the methodology express that the values of mRNA and protein expression were normalized using β-actin as a loading control, this could improve the text presented in the results.

Response 2: Thanks for your comment. In the study, we performed normalization using β-actin, and a brief normalized method had added the section of method as below:

4.4 Determination of protein levels by western blot

Proteins were extracted from cells in lysis buffer (Thermo Fisher Scientific, Waltham, MA, USA, #89901) containing proteinase and phosphatase inhibitor cocktails (Roche Diagnostics, Basel, Switzerland, #P0044-5ML). Equal amounts of protein (40 μg) were resolved by 10% sodium dodecyl sulfate-polyacrylamide gel electrophoresis and transferred to a polyvinylidene difluoride membrane (Millipore, Billerica, MA, USA). The membranes were blocked in 5% skimmed milk and incubated for 24 h at 4°C with the following primary antibodies: rabbit polyclonal anti-human EPB41L3 (1:2,000; Proteintech, Rosemont, IL, USA, #10719-I-AP), mouse monoclonal anti-human E-cadherin (1:1,000; Invitrogen, #33-4000), mouse monoclonal anti-human N-cadherin (1:1,000, Invitrogen, #33-3300), rabbit polyclonal anti-human collagen I (1:1,000; Abcam, Cambridge, MA, USA, #MA1-26771), mouse monoclonal anti-human-α-SMA (1:500; Abcam, #ab7817), mouse monoclonal anti-human fibronectin (1:1,000; Abcam, #ab6328), and mouse monoclonal anti-human β-actin (1:50,000; Sigma-Aldrich, St. Louis, MO, USA, #A1978). After washing several times with Tris-buffered saline containing Tween 20, the membranes were incubated with goat anti-rabbit (1:5,000; GenDEPOT, #SA007-500) or goat anti-mouse immunoglobulin G (IgG) horseradish peroxidase-conjugated secondary antibody (1:5,000, #SA001-500 and 1:100,000 for β-actin; GenDEPOT, #A0042-001). The membranes were analyzed using chemiluminescence [Thermo Fisher Scientific and Bio-Rad (Hercules, CA, USA)] with the ChemiDoc™ Touch Imaging System (Bio-Rad). The Protein expression were normalized using β-actin as a loading control.

Point 3: The authors demonstrate that overexpression of EPB41L3 in A549 and MRC5 cells by lenti-EPB41L3 transfection suppresses mRNA and protein expression associated with epithelial-mesenchymal transition (EMT) and fibroblast-myofibroblast transition (FMT). Whereas EPB41L3 silencing in A549 and MRC5 cells by EPB41L3 siRNA treatment increased the expression of mRNA and proteins associated with epithelial-mesenchymal transition (EMT) and fibroblast-myofibroblast transition (FMT). However, a major limitation of the study is that the functional role of EPB41L3 in IPF-related cellular processes associated with epithelial-mesenchymal transition (EMT) and fibroblast-myofibroblast transition (FMT), such as morphological changes, cell proliferation, migration, and invasion, was not fully assessed.

Response 3: We totally agree with the reviewer comment. Therefore, we added the limitation of our study as follows:

Our study has several limitations. First, control fibroblasts were obtained from normal portion of the resected cancer specimens. The gene expression profile of fibroblasts derived from lungs in which cancer developed may be different from that of fibroblasts derived from truly normal lungs. Secondly, we used the A549 and MRC-5 cell lines in the EMT and FMT studies instead of primary lung epithelial cells or fibroblasts. Thirdly, the EPB41L3 gene and protein levels were not measured using the sample such as bronchoalveolar lavage or lung tissues. To reveal the exact role of EPB41L3 in IPF, protein and gene levels of EPB41L3 should be evaluated in the lungs of the patients with IPF in terms of clinical manifestations by assessing the correlations of their levels with prognostic parameters, such as the long-term survival rate, in large number of patients. Additionally, changes of phenotypes including morphological changes, cell proliferation, migration, and invasion would be assessed in EPB41L3 overexpressed and knocked cell limes.

Round 2

Reviewer 3 Report

Dear authors, very kind for attending and making the changes to your manuscript, although I would have preferred that you mark the lines where you made the adjustments. However, the manuscript still requires minimal changes that do not involve much time. The references have the full names of the journals and are not abbreviated. In the supplementary material Figure S9, in the graph that shows the medians, locate the related test in the figure caption. Finally, the manuscript requires slight editing of the language.

Kind regards,

Minor editing of English

Author Response

Response to Reviewer 3 Comments

Point 1: Dear authors, very kind for attending and making the changes to your manuscript, although I would have preferred that you mark the lines where you made the adjustments. However, the manuscript still requires minimal changes that do not involve much time. The references have the full names of the journals and are not abbreviated. In the supplementary material Figure S9, in the graph that shows the medians, locate the related test in the figure caption. Finally, the manuscript requires slight editing of the language.

Response 1: Thank you for pointing that out. We changed the reference format to match the journal standard format.

Journal standard:

Journal references must cite the full title of the paper, page range or article number, and digital object identifier (DOI) where available. 

e.g.) 8. Bowman, C.M.; Landee, F.A.; Reslock, M.A. Chemically Oriented Storage and Retrieval System. 1. Storage and Verification of Structural Information. J. Chem. Doc. 19677, 43-47; DOI:10.1021/c160024a013.

We added the 5 revised references as follows:

  1. Richeldi, L.; Collard, H.R.; Jones, M.G. Idiopathic pulmonary fibrosis. Lancet 2017, 389, 1941-1952, doi:10.1016/s0140-6736(17)30866-8.
  2. Chambers, R.C.; Mercer, P.F. Mechanisms of alveolar epithelial injury, repair, and fibrosis. Ann Am Thorac Soc 2015, 12 Suppl 1, S16-20, doi:10.1513/AnnalsATS.201410-448MG.
  3. Kim, D.S.; Collard, H.R.; King Jr, T.E. Classification and natural history of the idiopathic interstitial pneumonias. Proceedings of the American Thoracic Society 2006, 3, 285-292.
  4. Moss, B.J.; Ryter, S.W.; Rosas, I.O. Pathogenic Mechanisms Underlying Idiopathic Pulmonary Fibrosis. Annu Rev Pathol 2022, 17, 515-546, doi:10.1146/annurev-pathol-042320-030240.
  5. Avci, E.; Sarvari, P.; Savai, R.; Seeger, W.; Pullamsetti, S.S. Epigenetic Mechanisms in Parenchymal Lung Diseases: Bystanders or Therapeutic Targets? Int J Mol Sci 2022, 23, doi:10.3390/ijms23010546.

Additionally, we added the statistical method in Figure S9 as below:

Revised figure legend:

Figure S9. Correlation analysis of EPB41L3 gene expression intensities with CpG DNA methylation levels of 12 fibroblasts.

(A) Comparison of EPB41L3 gene expression levels (GSE71351) between IPF (n = 8) and control fibroblasts (n = 4). The Mann–Whitney U test was performed to identify the statistical significance between IPF and control. Data are medians and quantiles. (B) CpG methylation levels of 21 loci on EPB41L3 were obtained from GSE107226. Correlation of the CpG methylation levels of 21 CpG loci on EPB41L3 with the gene expression level of the transcriptome chip was presented as correlation coefficient r (right Y-axis) and p-values as - log (left x-axis). The horizontal red line indicates a p-value of 0.05. TSS; transcription start site. UTR; upstream region.

Also, The English in this manuscript has been checked by at least two professional editors, both native speakers of English. For a certificate, please see:

http://www.textcheck.com/certificate/T3ibfR

Reviewer 4 Report

The authors present the revised version of their manuscript, the article has merit and presents some interesting data and improvements over the previous version, but I believe it is hampered by some critical weaknesses that need to be addressed.

One of the important limitations of the study is that the functional role of EPB41L3 in IPF-related cellular processes associated with epithelial-mesenchymal transition (EMT) and fibroblast-myofibroblast transition (FMT), such as morphological changes (microscopy), cell proliferation (MTT assays, OR CCK-8), migration (wound healing assay or transwell migration assay) and invasion (transwell invasion assay), was not fully assessed. These complementary studies would provide relevant information that would favor the importance of the presented study.  

Author Response

Response to Reviewer 4 Comments

Point 1: The authors present the revised version of their manuscript, the article has merit and presents some interesting data and improvements over the previous version, but I believe it is hampered by some critical weaknesses that need to be addressed.

One of the important limitations of the study is that the functional role of EPB41L3 in IPF-related cellular processes associated with epithelial-mesenchymal transition (EMT) and fibroblast-myofibroblast transition (FMT), such as morphological changes (microscopy), cell proliferation (MTT assays, OR CCK-8), migration (wound healing assay or transwell migration assay) and invasion (transwell invasion assay), was not fully assessed. These complementary studies would provide relevant information that would favor the importance of the presented study.  

Response 1: R4: We totally agree with the reviewer comment. Therefore, we added again the mainly limitation of our study as follows:

Previously sentence:

Our study has several limitations. First, control fibroblasts were obtained from normal portion of the resected cancer specimens. The gene expression profile of fibroblasts derived from lungs in which cancer developed may be different from that of fibroblasts derived from truly normal lungs. Secondly, we used the A549 and MRC-5 cell lines in the EMT and FMT studies instead of primary lung epithelial cells or fibroblasts. Thirdly, the EPB41L3 gene and protein levels were not measured using the sample such as bronchoalveolar lavage or lung tissues. To reveal the exact role of EPB41L3 in IPF, protein and gene levels of EPB41L3 should be evaluated in the lungs of the patients with IPF in terms of clinical manifestations by assessing the correlations of their levels with prognostic parameters, such as the long-term survival rate, in large number of patients. Additionally, changes of phenotypes including morphological changes, cell proliferation, migration, and invasion would be assessed in EPB41L3 overexpressed and knocked cell limes.

Revised sentence:

Our study has several limitations. First, control fibroblasts were obtained from normal portion of the resected cancer specimens. The gene expression profile of fibroblasts derived from lungs in which cancer developed may be different from that of fibroblasts derived from truly normal lungs. Secondly, we used the A549 and MRC-5 cell lines in the EMT and FMT studies instead of primary lung epithelial cells or fibroblasts. Thirdly, the EPB41L3 gene and protein levels were not measured using the sample such as bronchoalveolar lavage or lung tissues. To reveal the exact role of EPB41L3 in IPF, protein and gene levels of EPB41L3 should be evaluated in the lungs of the patients with IPF in terms of clinical manifestations by assessing the correlations of their levels with prognostic parameters, such as the long-term survival rate, in large number of patients. Additionally, one important limitation of our study is the lack of functional study of EPB41L3 in IPF-related cellular processes such as morphological changes, cell proliferation, migration and invasion. Therefore, further functional studies are needed using EPB41L3 overexpressed and knocked cell limes.
